# Cellular and Molecular Mechanisms of VSMC Phenotypic Switching in Type 2 Diabetes

**DOI:** 10.3390/cells14171365

**Published:** 2025-09-02

**Authors:** Shreya Gupta, Gilbert Hernandez, Priya Raman

**Affiliations:** 1Department of Biomedical Sciences, Northeast Ohio Medical University, Rootstown, OH 44272, USA; sgupta15@kent.edu (S.G.); ghernandez@neomed.edu (G.H.); 2School of Biomedical Sciences, Kent State University, Kent, OH 44242, USA

**Keywords:** VSMC phenotypic switching, hyperglycemia, obesity, insulin resistance, signaling pathways, transcriptional regulation, cellular reprogramming

## Abstract

Vascular smooth muscle cells (VSMCs) are a major cell type in the arterial wall responsible for regulating vascular homeostasis. Under physiological conditions, VSMCs reside in the medial layer of the arteries, express elevated levels of contractile proteins, regulate vascular tone, and provide mechanical strength and elasticity to the blood vessel. In response to obesity, hyperglycemia, and insulin resistance, critical pathogenic hallmarks of Type 2 diabetes (T2D), VSMCs undergo a phenotypic transformation, adopting new phenotypes with increased proliferative (synthetic), inflammatory (macrophage-like), or bone-like (osteogenic) properties. While crucial for normal repair and vascular adaptation, VSMC phenotypic plasticity is a key driver for the development and progression of macrovascular complications associated with T2D. Despite advances in lineage tracing and multi-omics profiling that have uncovered key molecular regulators of VSMC phenotypic switching in vasculopathy, our understanding of the cellular and molecular mechanisms underlying VSMC transformation into diseased phenotypes in T2D remains incomplete. This review will provide a holistic summary of research from the past 15 years, with a focus on the signaling pathways and transcriptional regulators that govern VSMC phenotypic transition in response to obesity, hyperglycemia, and insulin resistance. We examine the integrated molecular mechanisms that orchestrate VSMC fate reprogramming in T2D and highlight the dynamic interplay among diverse signaling and transcriptional networks. Emphasis is placed on how these interconnected pathways collectively influence VSMC behavior and contribute to the pathogenesis of T2D-associated atherosclerosis.

## 1. Introduction

Type 2 diabetes (T2D) is a multifactorial metabolic disorder defined by an array of pathogenic hallmarks that collectively drive disease initiation and progression. Central among these are obesity, insulin resistance, and hyperglycemia, the interplay of which forms a vicious cycle that triggers a cascade of metabolic disturbances [1]. T2D is a major risk factor for the development and progression of atherosclerosis, accounting for increased cardiovascular morbidity and mortality in affected individuals [2,3]. Despite the widespread use of statins as the gold standard lipid-lowering agent, statin-induced muscle pain, risks of hyperglycemia, and other adverse effects highlight the need for targeted interventions that address the distinct molecular mechanisms underlying T2D-induced atherosclerosis.

Atherosclerosis is a chronic inflammatory disease with multiple cell types residing in the vasculature, playing a critical role in disease progression. While damage and dysfunction of the innermost endothelial cell layer lining the arterial wall are classically viewed as hallmarks of atherosclerosis, other cell types, including monocytes, macrophages, vascular smooth muscle cells (VSMCs), and fibroblasts, engage in various stages of lesion formation [4,5]. Over the past two decades, state-of-the-art lineage tracing studies, combined with single-cell multi-omics profiling, have emphasized the contribution of VSMCs to atherosclerotic disease progression. VSMCs are pivotal in the regulation of vascular tone. In healthy vessels, VSMCs are confined to the medial layer of the vessel wall, where they acquire an organized cell arrangement of spindle-shaped morphology with unique contractile properties allowing for dynamic contraction and relaxation. These properties enable VSMCs to play a pivotal role in blood pressure regulation and the distribution of blood across various organs. However, in response to proatherogenic insult or stress (e.g., obesity, insulin resistance, hyperglycemia), VSMCs transform from contractile ‘quiescent’ differentiated state to synthetic ‘diseased’ de-differentiated phenotypes. Such phenotypic modulation of VSMC, referred to as phenotypic ‘switching’ or ‘transition’, is a key contributor to the initiation and advancement of atherosclerosis. While there is a wealth of literature on the mechanisms of VSMC plasticity in vascular biology, significant gaps persist in our understanding of the cellular and molecular mechanisms that regulate VSMC dysfunction in T2D. This review will summarize recent literature on the signaling and transcriptional mechanisms that mediate VSMC phenotypic changes driven by key pathophysiological hallmarks of T2D, such as obesity, hyperglycemia, and insulin resistance.

## 2. Obesity, Adipokines, and Atherosclerosis

Obesity is a chronic metabolic disorder that results from an imbalance between lipogenesis and lipolysis. The adipose tissue, initially viewed as an inert storage depot, is now well-recognized for its role as a source of metabolic fuel. Emergence of the adipose tissue as an endocrine organ initiated the discovery of a host of adipocyte-secreted factors capable of regulating glucose, lipid, and energy metabolism, inflammatory responses, and cell differentiation [6,7]. An imbalance in the secretion of adipose-derived factors, collectively termed ‘adipokines’, is a major contributing factor to obesity and related macrovascular complications [8]. Such adipose-derived factors can be broadly classified into pro-inflammatory or proatherogenic and anti-inflammatory or anti-atherogenic adipokines. Many of these adipokines influence the phenotype and function of VSMCs, leading to either beneficial or adverse effects on vascular function. In this review, we will discuss mechanisms of VSMC plasticity regulated by four important adipokines, namely, leptin, retinol-binding protein 4 (RBP4), adiponectin, and omentin-1.

### 2.1. Proatherogenic Adipokines and VSMC Dysfunction

#### 2.1.1. Leptin

Leptin, one of the most well-characterized adipokines, has significant implications in atherosclerotic complications associated with obesity. The long form of leptin receptor (Ob-Rb), the key mediator of leptin signaling, is abundantly expressed in VSMC. Previous studies document conflicting effects of leptin on the vasculature (reviewed in [9]). Specifically, leptin promotes VSMC growth and proliferation and elicits increased pressor responses mediated via activation of the renin-angiotensin system (RAAS) and release of endothelin, a potent vasoconstrictor [10,11]. On the other hand, leptin is known to induce nitric oxide (NO)-dependent arterial relaxation, and this effect is mediated via endothelial and neuronal nitric oxide synthase (NOS) [12]. The adverse vascular effects of leptin are primarily attributed to hyperleptinemia-induced reductions in arterial distensibility, mediated through activation of peripheral leptin receptors. In VSMC-specific leptin receptor knockout mice, the ability of leptin to impair acetylcholine-mediated vasorelaxation was significantly blunted [13], suggesting a VSMC-specific mechanism for hyperleptinemia-induced vascular dysfunction. Clinical and animal studies have highlighted a direct association between accelerated atherosclerosis and hyperleptinemia, a cardinal feature of obesity and metabolic syndrome (MetS) [14,15]. In obesity, the growth-promoting effects of leptin on VSMC lead to vascular remodeling, triggering atherosclerotic disease progression. Leptin stimulates VSMC migration, proliferation, and hypertrophy via activation of multiple signaling pathways, including JAK/STAT, PI3K/AKT, ERK1/2, p38-MAPK, mTOR, and RhoA/ROCK (reviewed in [9]). This results in altered actin dynamics contributing to hyperleptinemia-induced vascular dysfunction [16]. Previous studies have shown that increased matrix metalloproteinase (MMP2/9) expression and NADPH oxidase (NOX)-mediated generation of reactive oxygen species (ROS) controls leptin-induced VSMC proliferation and migration, triggering atherothrombotic events [17,18,19]. In high-fat diet-fed Wistar rats displaying MetS phenotype, perivascular adipose (PVAT)-derived leptin increased VSMC phenotypic switching via p38-MAPK activation, resulting in exacerbated vessel remodeling [20]. Consistent with these findings, we previously reported that exogenous leptin administration, at concentrations mimicking obesity, increases PCNA (proliferation marker), vimentin (SM synthetic marker), and pCREB (pro-mitogenic regulator of VSMC proliferation) expression, accompanied by reduced SM-MHC (SM contractile marker) expression in ApoE^−/−^ aortic vessels [21]. In obesity, elevated leptin has also been reported to induce osteoblastic differentiation of calcifying VSMC via ERK phosphorylation and PI3K/AKT activation, upregulating the RANKL-BMP4 axis [22]. Collectively, these findings suggest that aberrant leptin signaling contributes to functional deficits in VSMCs and highlight a regulatory role of leptin in VSMC phenotypic transformation, a central mechanism underlying obesity-associated macrovascular pathology. Key cellular and molecular mechanisms by which leptin regulates VSMC phenotypic transition are illustrated in Figure 1.

#### 2.1.2. Retinol Binding Protein 4 (RBP4)

RBP4 is a proatherogenic adipokine predominantly produced by the liver and adipose tissue. Circulating concentrations of RBP4 correlate with insulin resistance, obesity, and associated macrovascular complications [23,24]. Experimental and clinical studies have confirmed a positive association between upregulated RBP4 and the incidence of cardiovascular diseases, including coronary artery disease, stroke, and hypertension [25,26]. These data highlight RBP4 as an early predictor of cardiovascular disease, especially in the setting of T2D. In STZ-induced diabetic and high-glucose high-fat diet-induced diabetic atherosclerotic rats, elevated RBP4 expression in adipose tissues revealed a positive correlation with the atherogenic index of those animals. This was accompanied by increased tyrosine phosphorylation of JAK2 and STAT3 and elevated Bcl2 (regulator of apoptosis) and cyclin D1 (cell cycle regulator) expression [27]. Consistent with these results, Li et al. reported that RBP4 facilitates insulin-induced proliferation of rat aortic SMC, and this effect was mediated via ERK1/2 activation [28]. A recent study further demonstrated that in STZ-treated Wistar rats subjected to a high-glucose, high-fat diet feeding, intraperitoneal administration of RBP4 significantly increased RhoA, ROCK1, and mTOR expression [29]. Congruently, ectogenic RBP4 increased VSMC migration and proliferation while attenuating the expression of SM contractile (calponin, MYH11, α-SMA) and autophagy (LC3II/I and Beclin-1) markers in a mouse aortic SMC line (MOVAS) under high glucose conditions in vitro [29]. Together, the above findings suggest a potential role of RBP4 in VSMC phenotypic switching, contributing to macrovascular complications associated with obesity, possibly mediated via JAK2/STAT3 and RhoA/ROCK1 activation and inhibition of autophagy (summarized in Figure 2).

### 2.2. Anti-Atherogenic Adipokines and VSMC Dysfunction

#### 2.2.1. Adiponectin

Adiponectin is a multifaceted adipocyte-secreted protein with anti-inflammatory, vasoprotective, and insulin-sensitizing properties (reviewed in [30]). The role of adiponectin in the development of atherosclerosis has been somewhat conflicting. Although recombinant adenovirus-mediated expression of human adiponectin (*ADIPOQ*) was shown to attenuate atherosclerotic lesion formation in ApoE^−/−^ mice [31], a subsequent study demonstrated that *Adipoq* deficiency did not significantly alter lipid accumulation in the aortic roots of ApoE^−/−^ mice [32]. Similarly, LDLR^−/−^ mice overexpressing an adiponectin deletion mutant exhibited no significant difference in atherosclerotic lesion development compared to age- and sex-matched LDLR^−/−^ controls, despite showing improvements in plasma lipid profiles and lipid clearance [32]. While the role of adiponectin in atherosclerosis remains ambiguous due to its conflicting effects on lesion formation, existing literature suggests that VSMC-secreted endogenous adiponectin may be required to maintain SMC contractile gene expression. Specifically, knockdown of *ADIPOQ* in human coronary artery SMC (HCASMC) reduced SM contractile marker expression (ACTA2, Myh11, CNN1, and Tgln); this effect was replicated in cells isolated from *Adipoq*^−/−^ mice vs. wild-type VSMC [33]. Knockdown or overexpression studies further revealed an autocrine or paracrine effect of adiponectin on VSMC phenotype, and this effect was mediated via pro-differentiation transcriptional activators, such as Myocd and GATA6 [34]. Such pro-differentiation properties of adiponectin have been linked to activation of AMP kinase (AMPK) and inhibition of mammalian target of rapamycin complex 1 (mTORC1) [33]. Congruently, adiponectin inhibits VSMC migration and proliferation and induces cell apoptosis via reduced Ras-Raf-ERK1/2 signaling and upregulation of mitofusin, a mitochondrial fusion protein [35]. A previous study demonstrated that loss of adiponectin increases VSMC migration and proliferation via PI3K/AKT activation [33]. Moreover, although SM differentiation markers (ACTA2, Myh11, Tgln) were attenuated in *Adipoq*^−/−^ VSMCs, transcriptional activators of SM differentiation (Myocd, SRF) were upregulated together with reduced KLF4 (transcriptional repressor of SM contractile genes) protein expression. In addition, while loss of adiponectin reduced SRF nuclear translocation, no difference in nuclear Myocd or KLF4 expression could be detected under these conditions [33]. Collectively, these results support the idea that adiponectin stimulates VSMC contractile phenotype via activation of AMPK, an energy-sensing enzyme that responds to changes in energy levels by activating energy-producing pathways. Increased AMPK, in turn, reduces AKT and MAPK1 activity, resulting in reduced cyclin D1 expression, a known regulator of cell cycle progression [33]. These effects have been linked to reduced VSMC migration, proliferation, and oxidative stress concomitant to TGFβ-dependent regulation of extracellular matrix (ECM) in response to adiponectin. Taken together, the above findings implicate a regulatory role of AKT/MAPK1/TGFβ axis in adiponectin-mediated SMC differentiation. Adiponectin has also been reported to inhibit FOXO4 activity, modulated by AKT2-dependent phosphorylation of FOXO4, preventing FOXO4 nuclear translocation [36]. This, in turn, blocked the interaction of FOXO4, a transcriptional repressor of SM contractile genes, with Myocd, resulting in increased VSMC differentiation [36]. In Type 1 diabetic and non-diabetic individuals, low plasma adiponectin levels have been linked to the progression of coronary artery calcification, regardless of other cardiovascular risk factors [37]. Consistently, animal studies have reported increased vascular calcification and neointimal formation in *Adipoq^−/−^* mice via activation of the p38-MAPK pathway [38,39,40]. Adiponectin is also known to block VSMC transformation into osteoblast-like phenotypes, key players in the pathogenesis of vascular calcification; this effect is mediated via inhibition of mTOR and JAK2-STAT3 pathways [41,42]. Furthermore, attenuated RUNX2 and BMP2 expression mediate the inhibitory effect of adiponectin on osteoblast differentiation [38,41,42]. Together, the current literature lends strong support for pro-differentiation effects of adiponectin in VSMCs; these effects are mediated via regulation of AMPK/mTORC1, JAK2/STAT3, AKT/MAPK1/TGFβ, Ras-Raf-ERK1/2, and AKT2/FOXO4 pathways (summarized in Figure 3).

#### 2.2.2. Omentin-1

Omentin-1 is another adipocyte-secreted protein found in visceral stromal tissues. Clinical studies demonstrate an inverse relationship between serum omentin-1 levels and obesity, fasting insulin, and impaired glucose tolerance [43,44,45,46]. These data suggest a vasculoprotective role of omentin-1 in metabolic dysregulation. Elevated omentin-1 exerts a beneficial impact on cerebral ischemia, with atheroprotective effects mediated via its pleiotropic action on VSMC, endothelial cells, and macrophages [47,48]. The vasculoprotective effects of omentin-1 are mediated through diverse mechanisms, including suppression of VSMC and endothelial inflammation, inhibition of leukocyte attachment, downregulation of adhesion molecules, reduced macrophage infiltration and foam cell formation, and enhancement of NO-mediated vasodilation [49,50,51]. Increased levels of omentin-1 have been linked to reduced arterial stiffness, modulated via increased expression of elastin and fibronectin in VSMC [52,53]. Omentin-1 inhibits VSMC proliferation via AMPK/ERK pathway and prevents VSMC migration by inhibiting the NOX/ROS/p38/Hsp27 axis [47]. Incubation of human VSMCs with physiological concentrations of Omentin-1 was reported to attenuate growth factor-induced ERK phosphorylation, accompanied by increased AMPK activation [54]. Consistently, blockade of AMPK reversed omentin-mediated inhibition of VSMC growth and ERK phosphorylation. Similarly, transgenic mice overexpressing fat-specific human omentin-1 have shown reduced neointimal thickening and attenuated intimal hyperplasia in response to wire-induced vascular injury, and this effect was mediated via AMPK activation [54]. Omentin-1 also inhibits osteoblastic differentiation of calcifying VSMCs via PI3K/AKT-mediated stimulation of osteoprotegerin (OPG), a member of the TNF receptor superfamily, and inhibition of receptor activator of nuclear factor kappa-B ligand (RANKL) [47,53]. Both OPG and RANKL are regulators of bone resorption that play a vital role in bone metabolism. Additionally, incubation of calcifying VSMCs with exogenous human omentin-1 in vitro reduced VSMC calcification by suppressing RUNX2, collagen I, and osteocalcin expression [55]. Collectively, the current literature underscores a vasculoprotective role of omentin-1 mediated via regulation of AMPK/ERK, NOX/ROS, PI3K/AKT, and RUNX2/BMP2/RANKL pathways that drive VSMC phenotypic transition (summarized in Figure 4).

## 3. Hyperglycemia, Atherosclerosis, and VSMC Dysfunction

Chronic hyperglycemia, a classical hallmark of diabetes, has profound proatherogenic properties with a devastating impact on vascular function. Diabetic patients are predisposed to increased VSMC migration and proliferation, key events in the development and progression of atherosclerosis. Elevated concentrations of glucose alone are a major trigger for VSMC phenotypic changes, contributing to vascular remodeling. The following sections summarize studies that emphasize the cellular and molecular mechanisms underlying hyperglycemia-induced changes in VSMC fate and function.

### 3.1. High Glucose and VSMC Migration, Proliferation, and Inflammation

Migration, proliferation, and inflammation are central aspects of VSMC phenotypic alterations driven by hyperglycemia. Previous studies suggest an integrated role of protein kinase C (PKC), NOX-mediated ROS generation, JAK2 activation, and polyol pathway in VSMC hyperproliferative response to high glucose [56,57,58]. Rat aortic SMCs incubated with high glucose revealed a significant increase in mRNA and protein expression of proliferation markers (Cyclin D1, PCNA, and KLF4), coupled with reduced SM contractile marker expression (SM22) [59]. High glucose also increased the expression of migration-related markers such as MMP2 and MMP9 in VSMC cultures [59]. Congruently, high glucose induced ERK1/2 and AKT activation, signaling pathways associated with VSMC growth and proliferation. This was further accompanied by increased expression of HIF1α, a transcriptional regulator of VSMC proliferative growth factors [59].

Activation of JAK-STAT signaling is one of the key mechanisms responsible for high glucose-induced VSMC proliferation. Previous studies have reported that elevated glucose enhances Angiotensin II (Ang II)-induced JAK2 tyrosine phosphorylation, and this effect is mediated via direct interaction of JAK2 with AT1, an Ang II receptor in VSMC [60,61]. Ang II further increased tyrosine and serine phosphorylation of STAT1 and STAT3, transcriptional proteins downstream of JAK2. Immunoprecipitation of VSMC lysates with SHP1, a protein responsible for JAK2 tyrosine dephosphorylation, revealed that high glucose conditions deactivate SHP1, initiating prolonged JAK2 phosphorylation [60]. A subsequent study reported that activated STAT3 promotes SMC proliferation by suppressing Myocd-mediated activation of SM contractile genes [62]. Specifically, qPCR and immunoblotting showed a strong correlation between phosphorylated STAT3 expression and downregulation of SM contractile genes, including *MYOCD*, *SMA*, *SMMHC*, and *SM22*α, while upregulating pro-proliferative genes, such as Cyclin D1, *OPN*, and *PCNA*. Concurrently, STAT3 overexpression countered the ability of Myocd to upregulate SM contractile gene expression [62]. Thus, while Myocd increased the expression of contractile genes, its effects were profoundly diminished by STAT3, highlighting this transcriptional protein as a repressor of Myocd-mediated activation of SM contractile genes under hyperglycemic conditions.

A family of zinc finger transcription factors, known as kruppel-like factors (KLFs), plays a role in hyperglycemia-induced VSMC differentiation, proliferation, and inflammation. Previous studies have identified KLF5 as a key regulator of vascular inflammation in diabetes [63,64,65]. Briefly, VSMC derived from diabetic patients and high glucose-treated murine VSMC cultures showed augmented KLF5 and iNOS expression [66]. High glucose stimulated tyrosine nitration of KLF5, promoting its interaction with NFκB p50, and this in turn cooperatively increased the expression of inflammatory cytokines, such as TNFα and IL1β. In line with these findings, VSMC-specific KLF5 deletion significantly reduced inflammatory cytokine expression in the vascular tissues of diabetic mice [66]. These data support the regulatory role of KLF5 in inflammatory gene expression in diabetic vascular tissues. Emerging studies indicate that KLF5 acts as a transcriptional activator or repressor and regulates multiple physiological processes, including differentiation, development, and proliferation [67,68]. Specifically, KLF5 was identified as a target gene of microRNA 9 (miR9), alleviating VSMC de-differentiation under high glucose conditions [69]. VSMCs isolated from db/db mice, transfected with miR9 mimic, showed increased SM contractile marker expression coupled with reduced KLF5 and increased Myocd expression [69]. These results underscore the role of the miR9/KLF5/Myocd axis in hyperglycemia-induced VSMC de-differentiation and phenotypic switching. Another member of the KLF family known to mediate high glucose-induced VSMC proliferation and inflammation is KLF15. Several lines of evidence indicate that KLF15 regulates SMC response to vascular injury and is required to maintain the SMC contractile phenotype in vascular pathology [70,71]. It was previously reported that high glucose-treated murine aortic SMCs exhibit increased IL1β and TNFα mRNA expression, enhanced BrdU staining reflective of increased cellular proliferation, and reduced KLF15 mRNA and protein expression [72]. Consistently, KLF15 overexpression suppressed BrdU incorporation in glucose-stimulated VSMC. These findings support the role of KLF15 as a negative regulator of VSMC proliferation under hyperglycemic conditions.

FOXO1 is another transcription factor linked to VSMC inflammatory responses. Previous studies support a link between FOXO1 and apoptosis, stress, glucose metabolism, and diabetes [73,74,75]. FOXO1 mediates the transcriptional upregulation of IL1β expression [76]. In aortic vessels derived from high-fat-fed, STZ-treated diabetic mice, increased FOXO1, IL1β, and TNFα expression was noted compared to non-diabetic aortic vessels, postulating VSMC inflammation as a putative target of diabetic vasculopathy [77].

Hyperglycemia also triggers an upregulation of ROS-producing enzymes such as NOX and xanthine oxidase while downregulating the antioxidant enzymes. This imbalance disrupts cellular redox signaling and promotes oxidative stress. Hyperglycemia-induced imbalances in oxidative stress play a pivotal role in VSMC dysfunction in diabetes [78,79]. High glucose activates Toll-like receptor 4 (TLR4) signaling in VSMC, a key component of innate immunity activated in diabetes, resulting in increased ROS production. ROS in turn activates NFκB, resulting in increased proinflammatory cytokine expression. Briefly, high glucose-treated rat mesenteric VSMCs showed elevated TLR4 expression and signaling, while incubation with TLR4 inhibitor, CLI-095, reduced ROS and NFkB activation in high glucose-treated cells [80]. Analogously, streptozotocin-induced diabetic rats treated with CLI-095 in vivo revealed diminished ROS production accompanied by attenuated norepinephrine (NE)-induced contraction of mesenteric arteries [80]. These data support the role of TLR4 signaling in hyperglycemia-induced VSMC inflammatory responses. Hyperglycemia also increases the phosphorylation and membrane translocation of Rac1, p47phox, and p67phox subunits, triggering NOX-mediated ROS generation together with ERK1/2 and JNK1/2 activation in VSMC [81]. Notably, high glucose-induced cellular proliferation and increased ROS production were found to be contingent upon activation of p110α, the catalytic subunit of PI3K [81]. Excessive ROS accumulation directly stimulates VSMC proliferation, migration, and apoptosis, contributing to vascular remodeling and atherosclerosis. An earlier study demonstrated the role of the transient receptor potential cation channel subfamily M member 7 (TRPM7) in high glucose-induced VSMC proliferation and increased oxidative stress [82]. TRPM7 is a widely expressed Ca^2+^-permeable non-selective cation channel found in aortic VSMCs [83,84]. TRPM7 regulates VSMC growth, adhesion, contraction, cytoskeletal organization, migration, and differentiation to osteoblastic or proliferative phenotypes [85,86,87]. It was previously shown that knockdown of TRPM7 in rat aortic SMC partially blocks high glucose-induced SMC phenotypic switching and proliferation, and this effect was mediated via attenuated mitogen-activated protein kinase/extracellular signal-regulated kinase (ERK) kinase (MEK)-ERK signaling [82]. These data support a mechanistic role of the ROS-TRPM7-ERK1/2 axis as a driver of hyperglycemia-induced VSMC phenotypic alterations. A widely expressed protein known as the translocator protein (TSPO) has been implicated in diverse cellular processes, including proliferation, apoptosis, gene regulation, mitochondrial physiology, and immunomodulation [88,89,90,91]. Earlier work demonstrated that TSPO controls intracellular Ca^2+^ dynamics and redox transients in neuronal cytotoxicity [92]. Moreover, an interaction of TSPO with NOX links ROS to redox homeostasis. In an arterial injury rat model of diabetes, TSPO siRNA gene silencing diminished high glucose-induced VSMC proliferation and migration via the cGMP/PKG signaling pathway [93]. Taken together, the above findings delineate a pivotal role of mitochondrial homeostasis in VSMC dysfunction. Mitochondria, the powerhouse of ROS signaling, participate in vascular remodeling through diverse mechanisms. For instance, diminished mitochondrial biogenesis and mtDNA damage promote VSMC switching to a synthetic or osteogenic phenotype, contributing to vascular calcification [94]. It was previously reported that the transcriptional coactivator PGC-1α orchestrates mitochondrial biogenesis through activation of NRF1/2 and TFAM, ensuring adequate ATP production and redox homeostasis. Such PGC-1α-mediated mitochondrial biogenesis was shown to inhibit VSMC proliferation and senescence, thereby preserving the contractile phenotype [95]. Concurrently, dysregulated mitochondrial dynamics via increased DRP1-mediated fission and reduced MFN1/2- and OPA1-mediated fusion resulted in fragmented mitochondria and metabolic reprogramming, prompting VSMCs to adopt a synthetic phenotype [94,96,97]. Dysfunctional mitochondria also impair calcium signaling and apoptotic regulation, further destabilizing the contractile phenotype [98]. Mitochondria also play a multifaceted role in mitophagy, a specialized form of autophagy that removes damaged mitochondria to maintain cellular health [99]. This process is crucial in preventing ROS accumulation and maintaining energy homeostasis. Compelling evidence indicates that autophagy, a cellular process for the clearance and recycling of damaged cellular structures and organelles, acts as a gate-keeper of VSMC phenotypic switching [100,101]. While autophagy is important for maintaining the contractile phenotype of VSMCs, dysregulated autophagy may impose severe detrimental consequences on vascular function. Thus, impaired autophagy can lead to VSMC apoptosis and senescence, contributing to plaque instability. On the other hand, excessive or aberrant autophagy can trigger VSMC transformation into synthetic phenotypes, prompting vascular remodeling. It was previously reported that Notch receptor 3 (Notch3), a member of the Notch gene family, plays a vital role in preserving the phenotypic stability of VSMCs. Specifically, Xu et al. reported that in HASMC and HUASMC cultures in vitro, elevated glucose concentrations, mimicking a diabetic milieu, significantly increased Notch3 expression coupled with increased expression of autophagy markers such as LC3I/LC3II and Beclin1 [102]. Concurrently, siRNA-mediated gene silencing of Notch3 inhibited VSMC proliferation and expression of autophagy markers in high glucose-treated cells, suggesting a role of Notch3 in hyperglycemia-induced autophagy and VSMC proliferation. Additionally, in high glucose-treated VSMCs with Notch3 knockdown, overexpression of RANBP1, a regulatory protein of nuclear transport and cell cycle progression, rescued high glucose-induced changes in autophagy and cell viability [102]. These results suggest a regulatory role of the Notch/RANBP1 axis in autophagy and VSMC proliferation under hyperglycemia. Collectively, the above literature supports the concept that hyperglycemia-induced mitochondrial dysfunction facilitates VSMC phenotypic switching, implicating a mechanistic link between metabolic stress and vascular dysfunction in diabetes.

Glucose metabolism plays a critical role in VSMC phenotypic regulation. Synthetic VSMCs exhibit increased glycolytic flux. Previous work from our lab suggests a potential role of the hexosamine biosynthetic pathway, a minor branch of glycolysis, and downstream O-GlcNAc signaling, a post-translational modification (PTM), in hyperglycemia-induced VSMC proliferation [103,104]. A plethora of studies have indicated that elevated O-GlcNAcylation associates with upregulation of genes involved in inflammation [105,106], oxidative stress [107,108], matrix remodeling [109], and VSMC migration and proliferation (e.g., TSP1) [104]. O-GlcNAcylation competes with phosphorylation, and the crosstalk between these PTMs represents an important regulatory axis modulating cell behavior and function under physiological and pathological conditions [110]. More recently, we reported that in Western diet-fed hyperglycemic ApoE^−/−^ mice, lack of smooth muscle O-GlcNAc transferase (OGT), the key enzymatic regulator of O-GlcNAcylation, attenuates aortic root lipid burden and plaque area compared to hyperglycemic ApoE^−/−^ mice with intact OGT [111]. Notably, this atheroprotective effect of smooth muscle OGT deletion was accompanied by increased expression of SM contractile markers (ACTA2, LMOD1) and attenuated PCNA, pERK1/2, YY1, and SRF expression in the aortic vessels [111]. SRF is a transcription factor with a dual role in SMC phenotypic regulation that is contingent upon its binding partners [112,113,114]. The SRF binding to Myocd facilitates SMC differentiation, while its engagement with ETS transcription factors, including ELK1, drives VSMC proliferation and entry into the cell cycle [113,114,115]. Given that increased LMOD1 and ACTA2 expression was accompanied by attenuated YY1 and SRF expression in hyperglycemic ApoE^−/−^ aortic vessels lacking SMC-specific OGT, we postulate that the interaction of YY1 with SRF-dependent pathways may drive SMC de-differentiation under hyperglycemic conditions. Future studies, currently underway, are warranted to decipher the molecular link between O-GlcNAcylation and YY1/SRF-dependent pathways and delineate its role in VSMC fate switch in response to hyperglycemia. Aldose reductase (AR) is another glucose-metabolizing enzyme reported to mediate high glucose-stimulated VSMC proliferation [116]. AR is a cytosolic NADPH-dependent enzyme that functions in the polyol pathway of glucose metabolism, activated by high glucose. Earlier work has demonstrated that AR inhibition blocks high glucose-induced phosphorylation of retinoblastoma (Rb) protein and activation of E2F-1 [116]. This was accompanied by reduced phosphorylation of CDK-2 and attenuated expression of Cyclin D1, Cyclin E, CDK-4, c-myc, and PCNA (G1/S transition regulatory proteins). Collectively, the existing literature lends strong support to the idea that hyperglycemia functions as a metabolic cue modulating VSMC proliferation and cell cycle dynamics in diabetes. Figure 5A summarizes the diverse cellular and molecular cascades proposed to mediate high glucose-induced VSMC phenotypic transition.

### 3.2. High Glucose and Calcifying VSMC

Hyperglycemia is a known risk factor for vascular calcification, contributing to cardiovascular morbidity and mortality in diabetic patients. Vascular calcification is a pathological condition characterized by increased accumulation of calcium phosphate crystals in the vascular wall, resulting in stiffening or hardening of the blood vessels. Several mechanisms have been proposed to mediate vascular calcification, including increased ROS generation, endothelial dysfunction, chronic inflammation, and dysregulated calcium and phosphate homeostasis [117]. Chronically elevated glucose promotes VSMC transdifferentiation into osteoblast-like cells via activation of RUNX2 and BMP2/Msx-2/Sp7 axis [118]. Emerging evidence indicates that elevated plasma lactate levels, typically observed in diabetic patients, may contribute to vascular calcification by promoting pro-osteogenic signaling in VSMCs [119,120,121,122]. Lactate, a key by-product of glycolysis, serves as a substrate for ‘lactylation’, a newly discovered PTM linking cellular metabolism to gene regulation [123]. A recent study has revealed that under high glucose conditions, elevated lactate production promotes lactylation at the K18 site of histone H3 proteins in VSMC. This, in turn, activates JAK1/STAT3/RUNX2 signaling, promoting VSMC osteogenic transformation and accelerated arterial calcification in diabetes [124]. In addition, enhanced lactate levels modulate VSMC migration and proliferation, inflammatory response, and metabolic reprogramming, driving diabetic vascular pathology [119].

High glucose also regulates osteogenic transformation of VSMC via activation of PI3K/AKT and ERK1/2 signal transduction [125]. In thoracic aortic vessels derived from 4-month-old diabetic rats subjected to high-fat diet feeding and STZ treatment, a significant decrease in ACTA2 expression was noted, coupled with upregulated osteopontin expression, elevated alkaline phosphatase activity, and increased calcium content, reflective of vascular calcification. In VSMCs isolated from saphenous veins of patients undergoing coronary artery bypass grafting (CABG), high glucose in vitro stimulated the cytosolic translocation of high mobility group box 1 (HMGB1), a common ligand for the receptor for advanced glycation end products (RAGE) and toll-like receptors (TLRs) implicated in vascular calcification [126]. Notably, HMGB1 translocation was shown to be mediated via NOX- and PKC-dependent pathways. In addition, downregulation of HMGB1 abrogated high-glucose-induced calcification, accompanied by NFκB inactivation and reduced expression of BMP-2. These data imply a role of the HMGB1/NFkβ/BMP2 axis in hyperglycemia-induced vascular calcification. Low osteoprotegerin (OPG) levels following chronic hyperglycemia mediate osteo-inductive VSMC differentiation [127]. OPG, a member of the TNF-related family, is a component of the OPG/receptor activator of NFκB ligand (RANKL)/receptor activator of NFκB (RANK) [128]. Although earlier studies have suggested that OPG induces VSMC proliferation and fibrogenesis, whether OPG has a protective or detrimental role in the vasculature has been somewhat controversial. Kang et al. demonstrated that in rat VSMCs incubated with high glucose for 4 weeks, OPG mRNA and protein levels were significantly reduced, concomitant with increased beta-glycerophosphate and calcium deposition; these effects were observed in the absence of significant changes in the expression of RANKL, RANK, or TRAIL [127]. The authors further concluded that reduced OPG levels in response to chronic hyperglycemia may diminish OPG binding to RANKL or TRAIL, in turn accelerating vascular calcification. Another factor implicated in hyperglycemia-induced VSMC calcification is Sirtuin 1 (SIRT1), an NAD^+^-dependent deacetylase involved in hyperglycemia-induced VSMC transdifferentiation. Hyperglycemic conditions increased SA-β-galactosidase activity and p21 expression in murine VSMC, suggesting enhanced cellular senescence combined with attenuated SIRT1 expression [129]. Consistent with these data, activation of SIRT1 inhibited NFkB p65 acetylation, VSMC transdifferentiation, cellular senescence, and ROS production. Collectively, these findings support the concept that SIRT1 inhibits VSMC senescence and osteogenic differentiation via downregulation of NFκB activity, contributing to diminished vascular calcification. Figure 5B summarizes the signaling mechanisms that regulate VSMC osteogenic differentiation in response to high glucose.

## 4. Hyperinsulinemia, Atherosclerosis, and VSMC Dysfunction

Hyperinsulinemia, marked by dysregulated insulin secretion and/or insulin clearance, is commonly associated with obesity and metabolic dysregulation manifested in T2D [130]. Chronically elevated insulin levels lead to insulin resistance, profoundly diminishing the vasodilatory properties of insulin. This can result in adverse vascular outcomes, promoting atherosclerosis and hypertension. Multiple mechanisms are proposed to mediate atherosclerotic complications associated with chronic hyperinsulinemia and insulin resistance, namely, (i) endothelial cell-mediated synthesis of adhesion molecules triggering increased monocyte and macrophage recruitment, (ii) increased vascular inflammation, and (iii) increased VSMC growth, proliferation, and lipid uptake [131]. In the ensuing sections, we will review the existing literature on cellular and molecular mechanisms by which hyperinsulinemia promotes VSMC dysfunction in obesity, T2D, and related metabolic anomalies.

### Insulin and VSMC Proliferation, Inflammation, and Calcification

While some studies indicate that the vasoprotective effects of insulin are mediated via activation of PI3K-AKT-NO-mediated endothelial function [132,133,134], others suggest that insulin-mediated activation of the MAPK pathway stimulates VSMC proliferation [135,136,137]. Notably, the latter is preserved in insulin-resistant or insulin-deficient conditions, contributing to atherosclerosis. Consistent with the idea that insulin-mediated VSMC proliferation relies upon intact insulin signaling, VSMC-specific deletion of insulin receptor (IR) attenuated VSMC proliferation and wire injury-induced intimal hyperplasia in response to high-fat diet feeding [138]. The anti-inflammatory and mitogenic effects of IR signaling in VSMC can regulate both the severity and stability of atherosclerotic plaques in T2D and insulin resistance. Specifically, VSMC-specific loss of IR in ApoE^−/−^ mice increased inflammatory cytokines and adhesion molecules (e.g., ICAM1) and promoted VSMC transition into a macrophage-like phenotype [139]. Congruently, VSMCs isolated from ApoE^−/−^ mice with elevated ICAM1 and VCAM1 levels revealed diminished IR signaling compared to cells with lower levels of these inflammatory cytokines. These findings offer the first evidence that IR signaling regulates VSMC transformation into macrophage-like inflammatory phenotypes. VSMC-specific loss of IR in ApoE^−/−^ mice increased atherosclerotic lesions with reduced plaque stability [139]. These results support the concept that AKT-mediated insulin signaling in VSMCs acts as a suppressor of inflammation, extracellular matrix (ECM) turnover, and apoptosis, while promoting cellular proliferation. The selective loss of this signaling pathway may contribute to unstable plaque formation in cardio-metabolic diseases. Insulin-resistant ApoE^−/−^ mice lacking smooth muscle-specific IR showed increased expression of a matricellular protein, thrombospondin-1 (TSP1), in aortic vessels [139]. It is postulated that selective loss of AKT signaling reduces FOXO1 phosphorylation, facilitating its nuclear translocation, thereby enhancing TSP1 expression in insulin-resistant ApoE^−/−^ mice with SMC-specific IR deletion. TSP1 is significantly upregulated in atherosclerotic lesions, with increased localization in VSMC, in response to vascular injury and at sites of inflammation [103]. In VSMC, TSP1 regulates the expression of many inflammatory cytokines, including ICAM1, MMP2, and IL6, and exhibits potent proatherogenic and anti-angiogenic properties [103]. Accordingly, it is suggested that in insulin resistance, loss of AKT-mediated insulin signaling upregulates TSP1 expression in the vascular wall, contributing to VSMC transformation to inflammatory phenotypes. Importantly, we and others have demonstrated that hyperglycemia alone can upregulate TSP1 expression within the vascular wall [104,140,141]. Of note, in a murine model of combined metabolic syndrome (MetS) and atherosclerosis (KKAy^+/−^ApoE^−/−^), we recently reported that upregulated TSP1 expression directly associates with increased atherosclerotic lesion burden, accompanied by attenuated LMOD1 and SRF expression in the aortic vasculature [142]. Moreover, global TSP1 deletion increased SM contractile marker expression in the aortic vessels of MetS agouti KKAy^+/−^ mice compared with MetS mice with intact TSP1 [142]. These data accentuate a putative role of TSP1 in SMC de-differentiation in metabolic dysregulation, characterized by hyperglycemia and insulin resistance. Future investigations currently underway in our lab seek to define the specific role of smooth muscle-derived TSP1 in driving VSMC phenotypic transformation, with an emphasis on delineating the molecular mechanisms involved.

Insulin receptor substrate (IRS) plays a vital role in insulin signal transduction, affecting both positive and negative feedback mechanisms. IRS1 and IRS2 are two key proteins that mediate most insulin/insulin growth factor-1 effects, such as regulation of growth, differentiation, and metabolism. Reduced IRS function is linked to insulin resistance, and IRS1 is a critical regulator of VSMC differentiation. Both insulin resistance and hyperglycemia have been reported to downregulate IRS1 via increased ubiquitination and serine phosphorylation [143]. Gain- and loss-of-function studies indicate that insulin resistance or hyperglycemia-induced downregulation of IRS1 elevates KLF4 expression, a transcriptional repressor of the smooth muscle contractile phenotype. Increased KLF4 suppresses Myocd expression and disrupts its interaction with SRF, thereby promoting VSMC de-differentiation and proliferation. Additionally, IRS1 stabilizes p53, enhancing its nuclear association with KLF4; this interaction subsequently promotes Myocd-SRF binding, thereby contributing to increased VSMC differentiation [143].

Hyperinsulinemia or insulin resistance-induced VSMC migration and proliferation have also been linked to NOX-mediated oxidative stress. Prolonged insulin exposure (100 nM for 6 days) increases NOX expression and activity, elevates mitochondrial ROS production, and alters the expression of mitochondrial biogenesis-related genes in VSMCs.

Specifically, insulin treatment reduced mfn1 and mfn2 (mitochondrial fusion genes) while upregulating Drp1 (a mitochondrial fission gene), alongside increased Akt phosphorylation [144]. These molecular changes were accompanied by enhanced VSMC migration and proliferation. Notably, co-incubation of insulin with the NOX inhibitor diphenyleneiodonium normalizes these changes, suggesting a functional link between oxidative stress and mitochondrial dysfunction under conditions of hyperinsulinemia. These results support a potential crosstalk between NOX-derived ROS and mitochondrial dynamics in driving vascular dysfunction under conditions of insulin resistance.

Growing literature indicates that the transcription factor Msx2 can act as a molecular switch, favoring an osteoblast-like phenotype that may contribute to vascular calcification and loss of vascular elasticity [145,146,147]. In the context of how hyperinsulinemia promotes vascular calcification, earlier studies have shown that in insulin-resistant ob/ob mice, while Msx2 is important for osteochondrogenic differentiation of VSMC, Msx2 signaling alone may not be sufficient for vascular calcification [148]. Briefly, VSMC and aortic vessels isolated from insulin-resistant ob/ob mice revealed elevated Msx2, RUNX2, and alkaline phosphatase expression, and these effects were further enhanced following BMP2 treatment. These changes were accompanied by increased calcification and Smad1/5 phosphorylation, a mediator of VSMC calcification. Consistent with these results, Msx2 loss-of-function in ob/ob VSMC reduced BMP2-dependent osteochondrogenic differentiation compared to wild-type cells [148]. These data highlight a potential crosstalk between BMP2 and Msx2 pathways in VSMCs, regulating vascular calcification associated with insulin resistance. A summary of the cellular and molecular signaling pathways by which hyperinsulinemia regulates VSMC phenotypic changes is illustrated in Figure 6.

## 5. Converging Mechanisms of VSMC Plasticity in T2D

In obesity, hyperglycemia, and insulin-resistant states, VSMCs undergo phenotypic changes marked by increased proliferation, migration, cell cycle progression, and transformation into inflammatory and osteoblastic phenotypes. Given that T2D is driven by a complex interplay between obesity, hyperglycemia, and insulin resistance, understanding the molecular signals that govern their combined impact on VSMC fate and function is critical. The integrated signaling mechanisms by which these pathogenic hallmarks of T2D trigger VSMC dysfunction include PI3K/AKT, MAPK, AMPK/mTORC1, and BMP2 signaling. Extensive crosstalk among these signaling pathways orchestrates the phenotypic transition of VSMCs in T2D, contributing to disease-associated vascular remodeling. In addition, key transcriptional regulators that control VSMC proliferation, cell cycle progression, cell survival, and transformation into de-differentiated, inflammatory, and osteoblastic phenotypes in response to obesity, hyperglycemia, and insulin resistance, include KLF4, KLF5, FOXO1, STAT1/3, SRF, MYOCD, NFκB, and Msx2. Upstream of these convergent signaling and transcriptional pathways lies mitochondrial dysregulation, modulated in part by post-translational modifications (PTMs). Enhanced mitochondrial damage, increased autophagic activity, and disrupted mitochondrial dynamics, alongside elevated ROS production, collectively play a pivotal role in driving VSMC phenotypic switching in T2D. In addition, emerging evidence emphasizes the involvement of PTMs such as phosphorylation, O-GlcNAcylation, lactylation, ubiquitination, and acetylation in regulating VSMC plasticity in T2D, either directly by modulating key transcriptional regulators of VSMC transdifferentiation to diseased phenotypes or indirectly by modulating mitochondrial function and dynamics. Notably, an interplay between PTMs such as phosphorylation and O-GlcNAcylation modulates the activity of key kinases, including JAK, MAPK, AKT, and the energy-sensing enzyme AMPK, in diabetic VSMCs. These alterations in kinase activity may simultaneously activate or inhibit a broad spectrum of cytosolic and nuclear proteins, triggering the transcriptional upregulation of genes involved in inflammation, osteogenesis, proliferation, migration, cell survival, and apoptosis. Collectively, these changes contribute to VSMC phenotypic switching in T2D. Moreover, PTMs such as phosphorylation and O-GlcNAcylation often act in coordination or competition with ubiquitination and acetylation to fine-tune protein activity, stability, and localization in response to cellular signals. For instance, while ubiquitination governs the degradation of transcriptional regulators associated with the synthetic SMC phenotype, such as p53 and other chromatin-modifying transcription factors, acetylation and O-GlcNAcylation shield those proteins from ubiquitination, thereby stabilizing key transcriptional and signaling molecules involved in VSMC transdifferentiation. Additionally, these PTMs frequently work in concert to regulate nuclear import and export signals, ultimately influencing the subcellular localization of proteins, including protein–protein interactions that drive VSMC phenotypic transition. Recent lineage tracing and scRNA-seq studies have identified at least nine distinct VSMC-derived phenotypes, broadly categorized into intermediately and fully de-differentiated states [149]. This lends further support to the notion that an intricate network of transcriptional and epigenetic regulators of the synthetic, inflammatory, and osteogenic VSMC programs may be simultaneously activated, underscoring the complexity of VSMC phenotypic switching in T2D.

## 6. Conclusions and Future Perspectives

In conclusion, obesity, hyperglycemia, and insulin resistance, core pathophysiological hallmarks of T2D, regulate VSMC phenotypic switching via diverse signaling and transcriptional mechanisms. Although a wealth of data have identified converging signaling pathways, such as JAK/STAT, PI3K/AKT, MAPK, Rho/ROCK, AMPK/mTORC1, and NOX-ROS, that regulate VSMC fate and function in the context of obesity, hyperglycemia, and insulin resistance, there remains only a limited understanding of the integrated regulatory mechanisms by which these cardiovascular risk factors, co-existing in T2D, collectively drive VSMC phenotypic switching. For instance, while transcription factors such as FOXO1, RUNX2, and Msx2 have been implicated in VSMC transdifferentiation induced by both hyperglycemia and hyperinsulinemia, others like KLF5, KLF15 and YY1, proposed in the context of high glucose-induced VSMC de-differentiation, lack sufficient evidence supporting their involvement in VSMC fate change driven by hyperinsulinemia or obesity. Moreover, while an intricate network of molecular pathways is implicated in high glucose-induced VSMC signaling, the relative contribution of each cascade and their potential interactions, including synergistic or opposing effects on VSMC phenotype and function in the setting of co-existing metabolic anomalies (e.g., hyperglycemia, obesity, insulin resistance), is incompletely understood. Given the global prevalence of T2D and its associated vascular pathologies, an elucidation of the molecular mechanisms that govern VSMC fate transition into diseased phenotypes in response to complex metabolic or cardiovascular co-morbidities is of utmost importance. Current literature suggests that specific PTMs may lead to mitochondrial dysfunction, prompting VSMC phenotypic switching in T2D, contributing to vascular remodeling and accelerated atherosclerosis. Future studies are needed to uncover how these modifications fine-tune the balance between contractile and synthetic VSMC phenotypes in T2D. Additional studies are warranted to identify the downstream molecular target(s) that drive the transcriptional reprogramming of diseased VSMC phenotypes in T2D, including investigation of their putative interaction. Such information may lead to novel therapeutic avenues for the treatment and prevention of atherosclerotic complications associated with T2D.

## Figures and Tables

**Figure 1 cells-14-01365-f001:**
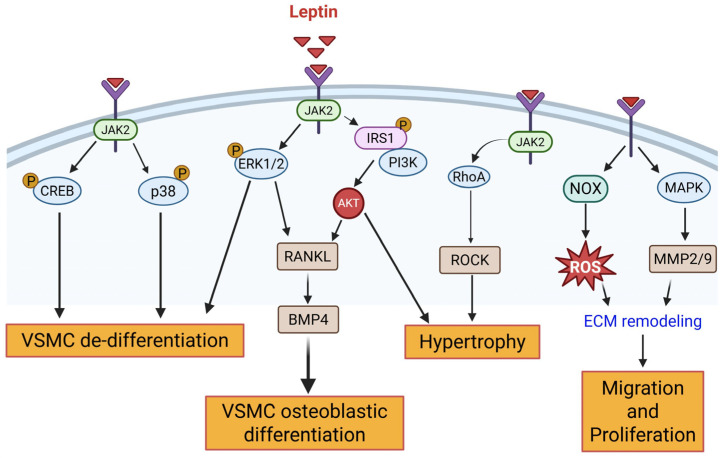
In obesity, hyperleptinemia induces activation of leptin receptors in VSMC, triggering multiple downstream signaling cascades contributing to vascular dysfunction. Abbreviations: VSMC: Vascular Smooth Muscle Cell; ECM: Extracellular Matrix; BMP4: Bone Morphogenetic Protein 4; IRS1: Insulin Receptor Substrate 1; NOX: NADPH Oxidase; CREB: cAMP Response Element-Binding Protein; JAK2: Janus Kinase 2; ERK1/2: Extracellular signal-Regulated Kinase 1/2; PI3K: Phosphoinositide 3-Kinase; ROCK: Rho-associated kinase; MAPK: Mitogen-Activated Protein Kinase; p38: p38 Mitogen-Activated Protein Kinase; RANKL: Receptor Activator of Nuclear Factor κB Ligand; MMP2/9: Matrix Metalloproteinase 2/9; ROS: Reactive Oxygen Species. Solid arrows denote activation pathways leading to downstream effects.

**Figure 2 cells-14-01365-f002:**
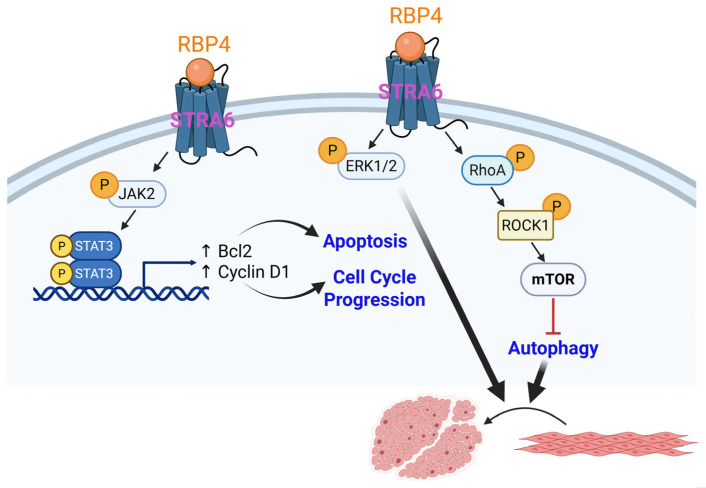
In obesity and insulin resistance, high levels of RBP4 via interaction with its receptor (STRA6) regulate VSMC de-differentiation via activation of JAK2/STAT3 and RhoA/ROCK1 pathways and inhibition of autophagy. Binding of RBP4 to STRA6 activates JAK2, which phosphorylates STAT3. Phosphorylated STAT3 translocates to the nucleus, promoting transcription of genes such as Bcl2 (anti-apoptotic) and cyclin D1 (promotes cell cycle progression). It also leads to phosphorylation of ERK1/2, contributing to the modulation of apoptosis and cell cycle progression. RBP4 engagement activates RhoA, subsequently phosphorylating ROCK1. This influences mTOR signaling, inhibiting autophagy, and triggers VSMC phenotypic switching. Abbreviations: JAK2: Janus Kinase 2; STAT3: Signal Transducer and Activator of Transcription 3; ERK1/2: Extracellular signal-Regulated Kinase 1/2; RhoA: Ras homolog family member A; ROCK1: Rho-associated kinase 1; mTOR: Mechanistic Target of Rapamycin; Bcl2: B-cell lymphoma 2; RBP4: Retinol binding protein 4. Solid arrows indicate activation pathways leading to downstream effects; T-bar (red) indicates inhibition; upward arrows indicate an increase; P in yellow circle indicates phosphorylation.

**Figure 3 cells-14-01365-f003:**
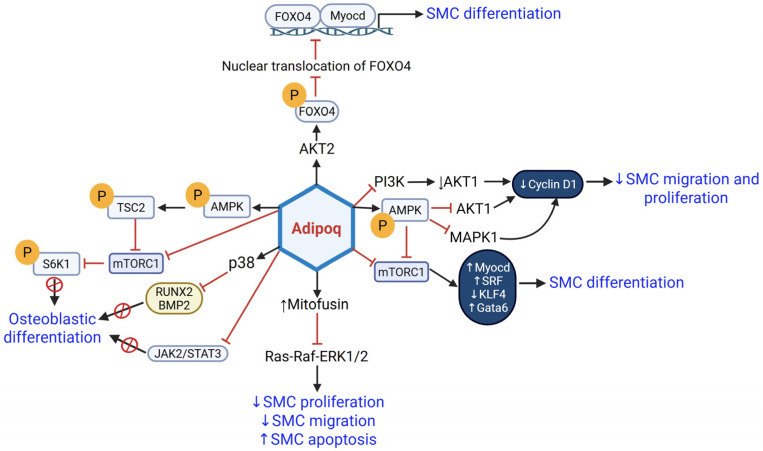
In obesity, low adiponectin levels block VSMC differentiation by activating JAK2/STAT3, Ras-Raf-ERK1/2, PI3K/AKT, mTORC1, RUNX2/BMP2, and FOXO4–Myocd interaction. Abbreviations: Adipoq: Adiponectin; JAK2: Janus Kinase 2; STAT3: Signal Transducer and Activator of Transcription 3; ERK1/2: Extracellular Signal-Regulated Kinase 1/2; PI3K: Phosphoinositide 3-Kinase; AKT: Protein Kinase B (PKB); mTORC1: Mechanistic Target of Rapamycin Complex 1; RUNX2: Runt-related transcription factor 2; BMP2: Bone Morphogenetic Protein 2; FOXO4: Forkhead box O4; Myocd: Myocardin; TSC2: Tuberous Sclerosis Complex 2; S6K1: Ribosomal Protein S6 Kinase Beta-1; AMPK: AMP-activated Protein Kinase; MAPK1: Mitogen-activated Protein Kinase 1. Solid arrows denote activation pathways leading to downstream effects. P in yellow circle denotes phosphorylation; red cross symbol or T-bar denotes inhibition; upward arrow indicates an increase; downward arrow indicates a decrease.

**Figure 4 cells-14-01365-f004:**
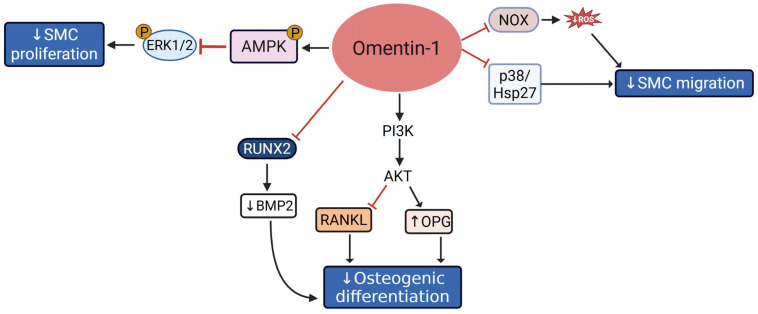
Vasculoprotective mechanisms of omentin-1 include AMPK-mediated inhibition of ERK1/2, PI3K/AKT-mediated regulation of RANKL-OPG balance, and blockade of NOX/ROS, p38/Hsp27 and RUNX2-mediated pathways. Abbreviations: ERK1/2: Extracellular Signal-Regulated Kinase 1/2; AMPK: AMP-activated Protein Kinase; PI3K: Phosphoinositide 3-Kinase; AKT: Protein Kinase B; NOX: NADPH Oxidase; ROS: Reactive Oxygen Species; p38: p38 Mitogen-Activated Protein Kinase; Hsp27: Heat Shock Protein 27; RUNX2: Runt-related transcription factor 2; BMP2: Bone Morphogenetic Protein 2; RANKL: Receptor Activator of Nuclear Factor κB Ligand; OPG: Osteoprotegerin. Solid arrow indicates activation pathways leading to downstream effects; T-bar (red) denotes inhibition; upward arrow indicates an increase; downward arrows indicate a decrease.

**Figure 5 cells-14-01365-f005:**
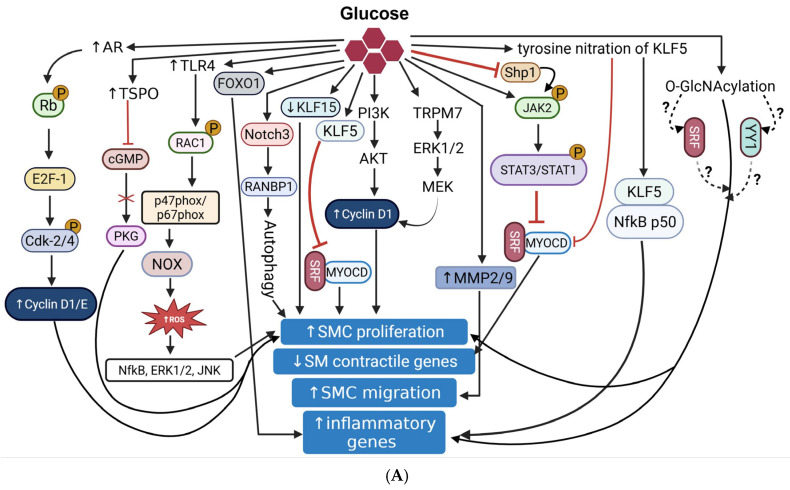
(**A**) Hyperglycemia regulates VSMC de-differentiation via activation of multiple cellular and molecular signaling cascades. Abbreviations: AR: Aldose Reductase; cGMP: Cyclic Guanosine Monophosphate; PKG: Protein Kinase G; NOX: NADPH Oxidase; ROS: Reactive Oxygen Species; NfkB: Nuclear Factor kappa-light-chain-enhancer of activated B cells; ERK1/2: Extracellular Signal-Regulated Kinase 1/2; JNK: c-Jun N-terminal kinase; FOXO1: Forkhead box O1; TLR4: Toll-Like Receptor 4; RANBP1: Ran Binding Protein 1; PI3K: Phosphoinositide 3-Kinase; AKT: Protein Kinase B; TRPM7: Transient Receptor Potential Melastatin 7; MEK: Mitogen-Activated Protein Kinase Kinase; KLF15: Krüppel-like factor 15; KLF5: Krüppel-like factor 5; SHP1: SH2-containing phosphatase 1; JAK2: Janus Kinase 2; STAT3/STAT1: Signal Transducer and Activator of Transcription 3/1; SRF: Serum Response Factor; MYOCD: Myocardin; MMP2/9: Matrix Metalloproteinase 2/9. Solid arrows denote activation pathways leading to downstream effects; dotted arrows denote putative activation pathways. T-bars (red) denote inhibition; upward arrow indicates an increase; downward arrow indicates a decrease. (**B**) Hyperglycemia mediates osteogenic transformation of VSMC via activation of RUNX2/BMP2/Msx1/Sp7, JAK1/STAT3, PI3K/AKT, NOX/PKC/HMGB1, OPG/RANKL and p21/SIRT/NFkB axis. Abbreviations: NOX: NADPH Oxidase; PKC: Protein Kinase C; HMGB1: High Mobility Group Box 1; JAK1: Janus Kinase 1; STAT3: Signal Transducer and Activator of Transcription 3; RUNX2: Runt-related transcription factor 2; BMP2: Bone Morphogenetic Protein 2; Msx2: Muscle Segment Homeobox 2; Sp7: Osterix; PI3K: Phosphoinositide 3-Kinase; AKT: Protein Kinase B; SIRT1: Sirtuin 1; p21: Cyclin-dependent kinase inhibitor 1A; ERK1/2: Extracellular Signal-Regulated Kinase 1/2; NfkB: Nuclear Factor kappa-light-chain-enhancer of activated B cells; Ac: Acetylation; OPG: Osteoprotegerin; RANKL: Receptor Activator of Nuclear Factor κB Ligand; TRAIL: TNF-related apoptosis-inducing ligand. Solid arrows denote activation pathways leading to downstream effects; T-bars (red) denote inhibition.

**Figure 6 cells-14-01365-f006:**
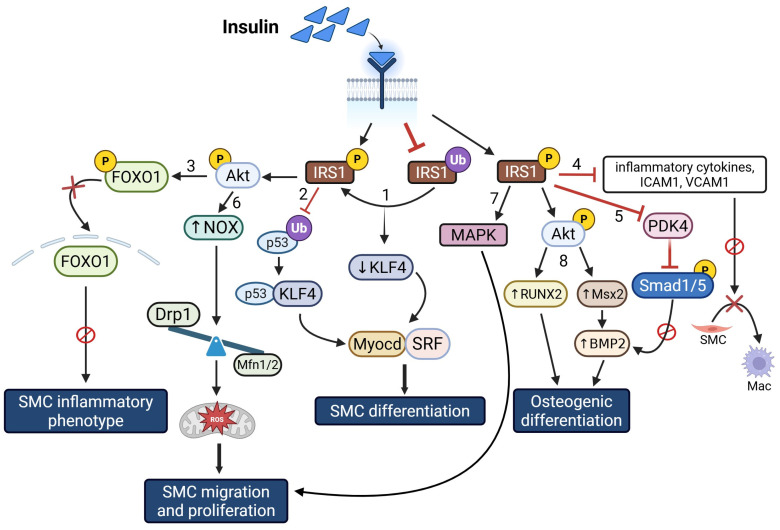
In insulin resistance, insulin receptor-IRS1-AKT-mediated signaling is impaired. While this blocks some downstream signaling pathways (1–5) triggering VSMC transdifferentiation, others (6–8) are preserved promoting SMC migration, proliferation and osteogenic differentiation. Abbreviations: IRS1: Insulin Receptor Substrate 1; AKT: Protein Kinase B; FOXO1: Forkhead box O1; NOX: NADPH Oxidase; p53: Tumor protein p53; KLF4: Krüppel-like factor 4; MAPK: Mitogen-Activated Protein Kinase; Drp1: Dynamin-related protein 1; Mfn1/2: Mitofusin 1/2; Myocd: Myocardin; Ub: Ubiquitination; RUNX2: Runt-related transcription factor 2; Msx2: Muscle segment homeobox 2; BMP2: Bone Morphogenetic Protein 2; Smad1/5: SMAD family member 1/5; ICAM1: Intercellular Adhesion Molecule 1; VCAM1: Vascular Cell Adhesion Molecule 1; SMC: Smooth Muscle Cell; Mac: Macrophage. Solid arrows denote activation pathways leading to downstream effects; T-bar (red) denotes inhibition; red cross symbols denote blocked pathways; upward arrows indicate an increase; downward arrows indicate a decrease.

## Data Availability

No new data were created or analyzed in this study.

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
