# Peer review of "Cellular and Molecular Mechanisms of VSMC Phenotypic Switching in Type 2 Diabetes"

_cells, 2025, doi:10.3390/cells14171365_

Round 1

Reviewer 1 Report

Comments and Suggestions for Authors

This important review summarizes the diverse signaling and transcriptional pathways that have emerged as important mediators of VSMC dysfunction in metabolic dysregulation. It acknowledges that despite the evidence of converging signaling pathways that control VSMC fate change and function in MetS, there is only limited understanding of the cumulative regulatory mechanisms by which obesity, hyperglycemia, and insulin resistance impact VSMC phenotypic switching.

Mitochondrial homeostasis is a significant factor in the pathogenesis of metabolic syndrome. While the manuscript summarizes key aspects of VSMC fate in MetS, it did not address mitochondrial homeostasis in VSMCs, which plays an important role in vascular remodeling. Emerging line of evidence suggests that mitochondria participate in vascular remodeling through multiple mechanisms. For example, peroxisome proliferator-activated receptor-γ coactivator-1α (PGC-1α)-mediated mitochondrial biogenesis inhibits VSMC from proliferation and senescence. Importantly, the imbalance between mitochondrial fusion and fission also regulates proliferation, migration and ultimately the phenotypic switching of VSMCs.

Therefore, addition of a paragraph to acknowledge the role of mitochondrial homeostasis in MetS and its potential role in VSMC fate could significantly enhance the quality of this review.

Reviewer 2 Report

Comments and Suggestions for Authors

This review article presents a comprehensive summary of the molecular mechanisms regulating vascular smooth muscle cell (VSMC) phenotypic switching in the context of metabolic dysfunctions such as obesity, hyperglycemia, and insulin resistance.

  1. The manuscript tends to list several signaling pathways in a descriptive manner without engaging in a deeper discussion of their interconnections. There is a notable lack of analysis regarding how these pathways might interact to regulate VSMC phenotypic switching, and the manuscript does not clearly identify which key molecular targets could be leveraged for therapeutic intervention in disease contexts.

It would also be helpful to include an abstract graphical summary that integrates the common signaling themes across different metabolic stressors to visually convey the central mechanisms of VSMC phenotypic switching.

  1. The discussion on epigenetic regulation (e.g., histone lactylation, O-GlcNAcylation) is a strong point. Expanding this section with more detail on how these post-translational modifications influence gene expression and VSMC fate would add significant value.
  2. Each figure should include a legend that fully explains all symbols, abbreviations, and directional arrows used.
  3. The title of the manuscript, "Cellular and Molecular Mechanisms of VSMC Phenotypic Switching in Metabolic Dysfunction," may be overly broad given the primary emphasis on diabetes-related mechanisms throughout the text with many of the cited studies centered on hyperglycemia and diabetic models.

Round 2

Reviewer 2 Report

Comments and Suggestions for Authors

1. The revised version demonstrates significant effort to address some of the prior concerns, but several important gaps remain. While the manuscript now provides more comprehensive descriptions of individual signaling pathways and metabolic mediators, it still lacks integration across pathways and fails to fully contextualize their collective roles in regulating VSMC phenotypic switching. An abstract graphical summary is highly recommended.
